



# Stratospheric Temperature Measurements from NanoSatellite Observations of Stellar Occultation Bending

Dana L. McGuffin[1], Philip J. Cameron-Smith[1], Matthew A. Horsley[1], Brian J. Bauman[1], Wim De Vries[1], Denis Healy[2], Alex Pertica[1], Chris Shaffer[2], and Lance M. Simms[1]

[1]Physical and Life Sciences, Lawrence Livermore National Laboratory, Livermore, CA, United States
[2]Terran Orbital, Irvine, CA, United States

**Correspondence:** D.L. McGuffin dana.lynn.mcguffin@gmail.com

**Abstract.** Stellar occultation observations from space can probe the stratosphere and mesosphere at a fine vertical scale around the globe unlike other measurement techniques like radiosondes, aircraft, and radio occultation. We imaged the refractive bending angle of a star centroid for a series of occultations by the atmosphere. Atmospheric refractivity, density, and then temperature are retrieved from the bending observations with the Abel transformation and Edlén's law, the hydrostatic equation, and the ideal gas law. The retrieval technique is applied to data collected by two nanosatellites operated by Terran Orbital. Measurements were primarily taken by the GEOStare SV2 mission, with a dedicated imaging telescope, supplemented with images captured by spacecraft bus sensors, namely the star trackers on other Terran Orbital missions. The bending angle noise floor is 10 arcseconds and 30 arcseconds for the star tracker and GEOStare SV2 data, respectively. The most significant sources of uncertainty are due to centroiding errors due to the fairly low-resolution stellar images and telescope pointing knowledge derived from noisy satellite attitude sensors. The former mainly affects the star tracker data, while the latter limits the GEOStare SV2 accuracy with both providing low vertical resolution. This translates to a temperature profile retrieval up to roughly 20 km for both star tracker and GEOStare SV2 datasets. In preparation of an upcoming 2023 mission designed to correct these deficiencies, SOHIP, we simulated bending angle measurements with varying magnitudes of error. The expected maximum altitude of retrieved temperature is 41 km on average for these simulated measurements with a noise floor of 0.39 arcseconds. Our work highlights the capabilities of stellar occultation observations from nanosatellites for atmospheric sounding. Future work will investigate high frequency observations of atmospheric gravity waves and turbulence, mitigating the major uncertainties observed in these datasets.

## 1 Introduction

Occultation measurement techniques perform atmospheric sounding by observing the perturbation in a signal that passes through the atmosphere. A signal source on the opposite side of the Earth from a satellite is tracked as the line of sight between them passes deeper into the atmosphere until the Earth limb obscures the signal completeley. The signal utilized for Radio Occultation (RO) and Stellar Occultation (SO) are radio waves from Global Positioning System (GPS) satellites and optical waves from starlight, respectively. SO was first described by Jones et al. (1962) while Yunck et al. (1988) first described RO.



Since 1995 many RO missions have retrieved atmospheric density and temperature accurately in the upper troposphere and
lower stratosphere up to 30-35 km (Hajj et al., 2002), which are utilized for research on diverse topics like numerical weather
prediction and deep convection in the tropics (Anthes, 2011; Scherllin-Pirscher et al., 2021). SO has been utilized to study
Earth's atmosphere via refraction (White et al., 1983; Yee et al., 2002; Sofieva et al., 2003). In 2002, the first instrument
dedicated to SO was launched on the ENVISAT (ENVIronmental SATellite) platform. The instrumentation utilizes absorptive
SO techniques to observe $O_3$ trends (Bertaux et al., 2010).

Both RO and SO are intrinsically relative measurements since they observe the change in a signal with and without at-
tenuation from the atmosphere to determine atmospheric properties. This feature makes these techniques perfect candidates
for long-term observations since there is minimal measurement drift. Additionally, both techniques utilize an already existing
signal (starlight and GPS radio waves for SO and RO, respectively) and only require the signal detector or receiver onboard the
payload. The biggest difference between the two occultation methods is the wavelength of the signal. The small wavelength
of optical waves compared to microwaves allows SO to interrogate small spatial scales and observe atmospheric turbulence
(Gurvich and Kan, 2003a, b; Sofieva et al., 2007). This work focuses on refractive SO because of the potential to measure both
temperature and turbulence.

Previous SO Earth observations were performed on large satellites: Orbiting Astronomical Observatory (OAO-2), Mir satel-
lite, Global Ozone by Stellar Occultation (GOMOS) on ENVISAT, and Ultraviolet and Visible Imagers and Spectrometers
(UVISI) on the Midcourse Space Experiment (MSX) (Hays and Roble, 1973; Gurvich and Kan, 2003a; Bertaux et al., 2010;
Yee et al., 2002). As noted above, SO only requires a signal detector, which does not require a large or complex platform.
Additionally, refractive SO requires high quality camera optics compared to the spectrometer necessary for absorptive SO, and
hence a much smaller platform can be used for refractive SO.

Developments in spacecraft technology since 1997 have spurred an exponential growth in launch rates of "nanosatellites"
with 1-10 kg mass (Janson, 2020). Nanosatellites are also at least 100 times cheaper and much quicker to launch than typical
full-scale satellites (Woellert et al., 2011). We can launch multiple nanosatellites for the same cost of a full-scale satellite and
observe the atmosphere with a higher temporal resolution. Alternatively, multiple nanosatellites can perform measurements at
various angles enabling advanced retrieval techniques like tomography. The trade-off for the nanosatellite low cost is a satellite
with decreased stability. However, we can mitigate the nanosatellite instability by utilizing two cameras: a wide-view camera
to image reference stars and a narrow-view camera to observe occulting stars. This paper describes the refractive SO technique
utilized to perform atmospheric sounding in Section 2. Then, two datasets from operational nanosatellites are presented and
analyzed for a proof-of-concept in Section 3. Sounding observations from a star tracker are presented in Section 3.1. Observa-
tions from a high-resolution telescope on-board the GEOStare SV2 nanosatellite are shown in Section 3.2. Section 4 analyzes
the retrieval error with varying levels of simulated instrument error and compares the results to the observed error from the star
tracker and GEOStare SV2 data. Finally, Section 6 discusses the improvements necessary to make the SO technique viable for
observing stratospheric properties.



## 2 Description of Stellar Occultation Sounding Technique

Remote atmospheric sounding with refractive stellar occultation (SO) utilizes images of a star as it sets below the Earth's horizon to determine the atmospheric temperature vertical profile. Section 2.1 describes how the ray "bends" due to the refractive index profile and Section 2.2.2 describes the method(s) to determine the observed refractive bending angle vertical profile from the satellite images.

The observed bending at each level is due to changes in the refractive index along the line of sight from the satellite to the star, so an observation at a particular altitude is an integrated measurement from the ray perigee to the top of the atmosphere. Thus, the refractive index vertical profile can be retrieved from the observed overall bending angle with an "onion-peeling" inversion approach. Section 2.3 describes the retrieval technique, the equations to determine atmospheric state based on the retrieved refractive index profile, and the effect of instrument error on the retrieved profile. The atmospheric properties determined from these measurements include density, pressure, and temperature.

### 2.1 Atmospheric Refraction Forward Model

The SO atmospheric sounding technique observes the atmosphere's effect on light generated by a star. The atmosphere appears to "bend" light that passes through due to the atmospheric gradient in refractive index. The atmospheric vertical profile of refractive index $n(z)$ varies with atmospheric density $\rho(z)$ based on the surface density $\rho_0$ for a wavelength $\lambda$ (Edlén, 1966):

$$n(z) = 1 + C(\lambda)\frac{\rho(z)}{\rho_0} \tag{1}$$

where $C(\lambda)$ is a dispersion factor, describing the variation of refractive index with wavelength (Barrell and Sears, 1939).

$$C(\lambda) = 10^{-8}\left(8342.13 + \frac{2.406 \cdot 10^6}{130 - \lambda^{-2}} + \frac{15997}{38.9 - \lambda^{-2}}\right) \tag{2}$$

The refractive index profile can be written in terms of atmospheric pressure $P(z)$ and temperature $T(z)$ profiles based on the Ideal Gas law with gravitational acceleration $g$ and the specific gas constant for dry air $R_{air}$. Based on the hydrostatic equation and a known temperature gradient $\mathrm{d}T/\mathrm{d}z$, the refractive index vertical gradient is differentiated from Eq (1):

$$\frac{\mathrm{d}n}{\mathrm{d}z} = -C(\lambda)\frac{T_0 P(z)}{P_0 T(z)^2}\left(\frac{g}{R_{air}} + \frac{\mathrm{d}T}{\mathrm{d}z}\right). \tag{3}$$

The refractive index profile and its gradient are used in the Eikonal equation to model the path of a ray passing through the atmosphere. A ray with position $\boldsymbol{x} = (x, y, z)$ in Cartesian coordinates along its path $s$ is traced with:

$$\frac{\mathrm{d}^2\boldsymbol{x}}{\mathrm{d}\tau^2} = n\nabla n \tag{4}$$

where $\mathrm{d}\tau = \mathrm{d}s/n$ (Zou et al., 1999). The ray tracing equation above is integrated in 1-D assuming that the atmospheric refractive index vertical profile does not vary with latitude or longitude. Tracing the ray starts at the orbiting satellite telescope located 430 km above the Earth surface. The ray direction is initialized from the satellite position with a vector pointing toward





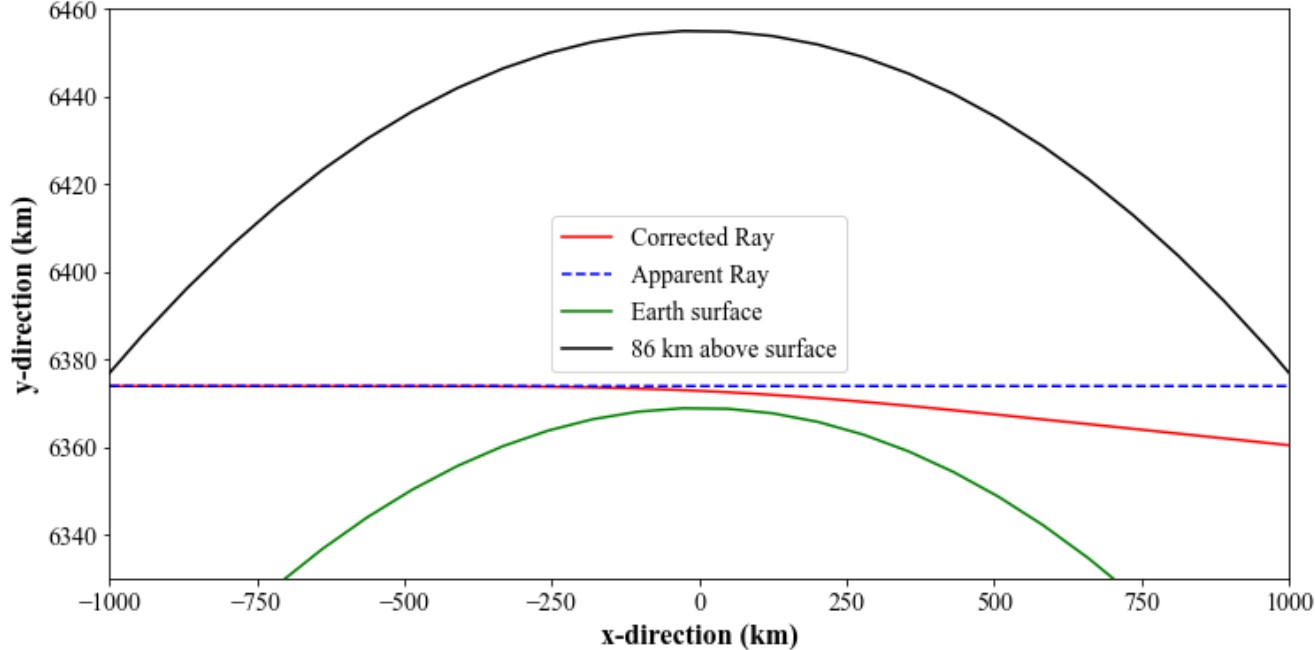

**Figure 1.** Example of ray trajectory through an atmospheric profile from MERRA2. The black and green lines do not look circular because of the different x- and y-scales. Ray-tracing computes both the ray perigee and the bending angle for an apparent ray. Here, an apparent ray perigee of 5 km results in a corrected ray perigee of 3.77 km with 2,914 arcseconds bending angle.

its apparent ray perigee, i.e. the altitude of the point closest to the surface. Then, the corrected ray path is calculated by integrating the Eikonal equation (4) based on a vertial profile of temperature and pressure. The atmospheric profile utilized here is from the Modern-Era Retrospective analysis for Research and Applications, Version 2 (MERRA2) reanalysis dataset (Global Modeling and Assimilation Office (GMAO), 2015b). Figure 1 shows the actual ray path due to refraction results in a corrected

ray perigee of 3.77 km for an apparent ray with a line of sight perigee of 5 km. The angle between the corrected ray and the line of sight, normal to the Earth surface, is the refractive bending angle, which is 2900 arcseconds in Figure 1. Top of the atmosphere, above which the refractive bending angle is negligible, is assumed to be 86 km corresponding to the maximum altitude predicted by MERRA2.

## 2.2 Refractive Bending Observations

Observing the bending angle is accomplished by commanding a camera to capture images in inertial pointing mode with the telescope pointed at the star of interest. Any motion of the star relative to its initial position is attributed to changes in the refractive index of the atmosphere, and we track the star position to calculate the vertical profile of observed refractive bending angles. First, we determine the star centroid position on each image in terms of pixels on the field of view during an occultation session. Next, we find the location of the field of view in the sky using stars above the atmosphere, so the star position on each





image can be compared on an absolute reference frame. The bending angle of the star of interest in each image is determined

from the star's centroid change in position on the absolute reference frame, and we calculate the apparent and corrected ray

perigees.

### 2.2.1 Centroid Position

Diffraction of a stellar point source in the telescope is captured on the detector as a point spread function (PSF). Irregular

deviations in the optical surfaces and atmospheric seeing conditions may result in a Moffat PSF (Moffat, 1969). In this work,

we find the centroid location on the image $(x_0, y_0)$ by fitting a PSF to the observed star centroid within a square region of 20

by 20 pixels. The two functions we used are the 2D Gaussian ($f_g$) and Moffat ($f_m$) distributions.

$$f_g(x,y) = A_g \exp\left\{ -\frac{(x-x_0)^2}{\sigma_x^2} - \frac{(y-y_0)^2}{\sigma_y^2} \right\} \tag{5}$$

$$f_m(x,y) = A_m \left( 1 + \frac{(x-x_0)^2 + (y-y_0)^2}{B^2} \right)^{-\beta} \tag{6}$$

Both PSFs require three parameters in addition to the centroid location: a parameter for the peak intensity ($A_g$ or $A_m$) and two

parameters describing the distribution shape. The 2D Gaussian includes $\sigma_x$ and $\sigma_y$ for the x- and y-width in pixels, respectively.

The Moffat distribution includes a width parameter $B$ and a negative exponent, $-\beta$, that determine the PSF shape. The Moffat

distribution is similar in shape to a Gaussian with a tail or "wing" that becomes more prominent as $\beta$ decreases with typical

values below 5 Trujillo et al. (2001).

### 2.2.2 Bending Angle

In inertial pointing mode, the camera keeps pointing at the same starfield as the payload orbits. However, spacecraft pointing is

not perfectly steady due to drift and jitter, and this motion must be accounted for to determine the relative change in the centroid

as the starlight passes deeper into the atmosphere. The camera view is defined by its boresight, or center of the image specified

by its coordinates in Right Ascension (RA) and Declination (DEC), along with camera rotation around the boresight. These

parameters can be calculated at the time of exposure by finding an astrometric solution for the starfield above the atmosphere,

or they can be estimated from the payload attitude and position control system. The World Coordinate System (WCS) defined

by these parameters is utilized to translate the centroid location on each image to a stabilized reference frame.

The stabilized reference frame is chosen as the average WCS among all images with boresight ray perigees above 100 km.

The actual coordinate of the star is determined by the average world coordinate ($RA^*$,$DEC^*$) among all images with boresight

ray perigees above 100 km. Refractive bending angle of each image is determined by calculating the euclidean distance from

the reference coordinate's pixel location on the image to the observed centroid pixel location. The detector plate scale (pixels

per radian) is used to convert the pixel distance to the refractive bending angle.



### 2.2.3 Ray Perigee

The apparent and corrected ray perigees must be calculated for each image based on the information known about the telescope's pointing and the observed star position since the star is not necessarily in the center of the image. We utilize two approaches to determine the apparent ray perigee of each measurement with the Ansys Systems Tool Kit (STK) depending on the data available. STK is a software that simulates platforms and payloads including satellites based on the known satellite location and orbit. As explained next, we use a direct approach if accurate information for the camera WCS is available at each exposure time. Otherwise a rotated approach is used.

*Direct.* The direct approach utilizes the STK software to draw a vector directly from the satellite position to the observed centroid coordinate. The coordinate is determined from the image WCS and the centroid location on the image. The software determines the point along the vector closest to the Earth surface and returns the altitude, latitude, and longitude of the ray perigee. This method is best suited for scenarios with high certainty on the image WCS since it defines the line of sight vector.

*Rotated.* The rotated approach utilizes the star reference coordinate and bending angle instead of the the image WCS. Although the WCS is used in calculating the bending angle as described above, it is not directly used by STK so that the ray perigee is less sensitive to errors in the image WCS. Here, a reference vector is drawn from the satellite to the occulted star's actual coordinate. This vector is rotated normal to the Earth surface by the observed refractive bending angle. Then, the ray perigee and its latitude and longitude are accessed from the rotated vector's point closest to the Earth surface.

## 2.3 Retrieval Technique

The observations result in a vertical profile of refractive bending angle. To investigate the underlying atmospheric state, these observations must be inverted for the refractive index vertical profile. Then, the inverse of Eq. (1) is applied to estimate atmospheric density.

### 2.3.1 Bending Angle Inversion

The refractive index profile can be retrieved by applying the inverse Abel transform to the bending angles $\alpha$ observed above a particular ray perigee ($z$) (Sofieva and Kyrölä, 2004). The Abel integral is relative to an impact parameter $p = n(z)r$ where $r$ is the distance from the Earth center, $r = z + r_E$ assuming the Earth's radius $r_E = 6371$ km. However, since the atmospheric refractive index, $n(z)$, is between 1 and 1.0003, $n$ can be neglected so that $p \approx z + r_E$. Applying the Abel inversion to the bending angle profile estimates the refractive index as a function of $r$:

$$\ln(n(r)) = \frac{1}{\pi} \int_{r}^{\infty} \frac{\alpha(a)}{\sqrt{a^2 - r^2}} \mathrm{d}a \tag{7}$$

where $a$ corresponds to the distance from Earth's center.



### 2.3.2 Temperature Calculation

Since atmospheric density is related to the refractive index, as described in Eq. (1), we estimate the atmospheric density profile

from the retrieved refractive index profile based on Edlén's equation with $C(\lambda)$ described by Eq (2).

$$\rho(z) = \frac{n(z) - 1}{C(\lambda)} \rho_0 \tag{8}$$

The hydrostatic equation relates atmospheric pressure and density profiles so the pressure vertical profile can be estimated from the atmospheric density profile with Eq (9). Then, the Ideal gas law determines atmospheric temperature based on the pressure and density, so the temperature vertical profile is estimated from the retrieved refractive index using Eq (10).

$$P(h) = \int_h^\infty \rho(z) \frac{g\, r_E^2}{(r_E + z)^2} \mathrm{d}z \tag{9}$$

$$T(h) = \frac{P(h)}{R_{air}\rho(h)} \tag{10}$$

## 3 Satellite Measurements

Refractive stellar occultation measurements were collected with two different nanosatellite instruments in orbit. Both satellites were launched and operated by Terran Orbital. Observations collected by a star tracker onboard a nanosatellite between

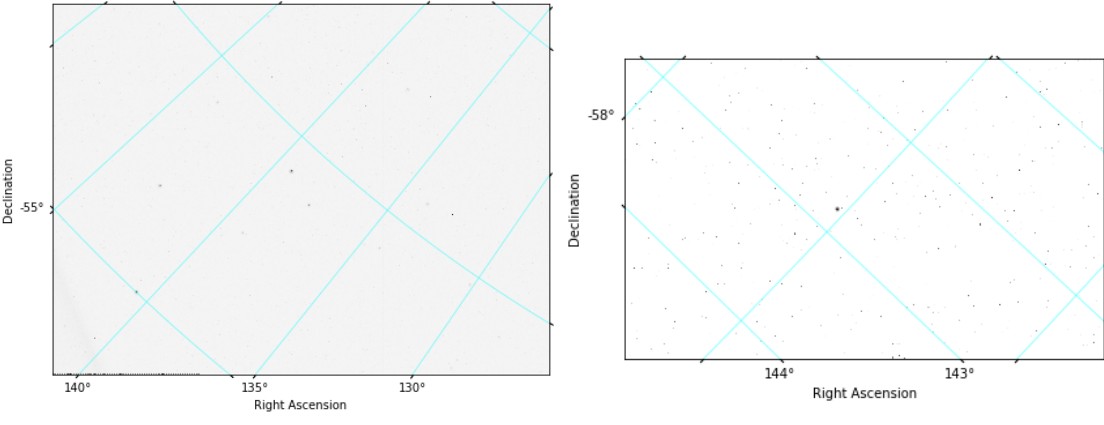

a: Starfield from ST          b: Starfield from GEOStare SV2

**Figure 2.** The same starfield imaged by ST and GEOStare SV2.





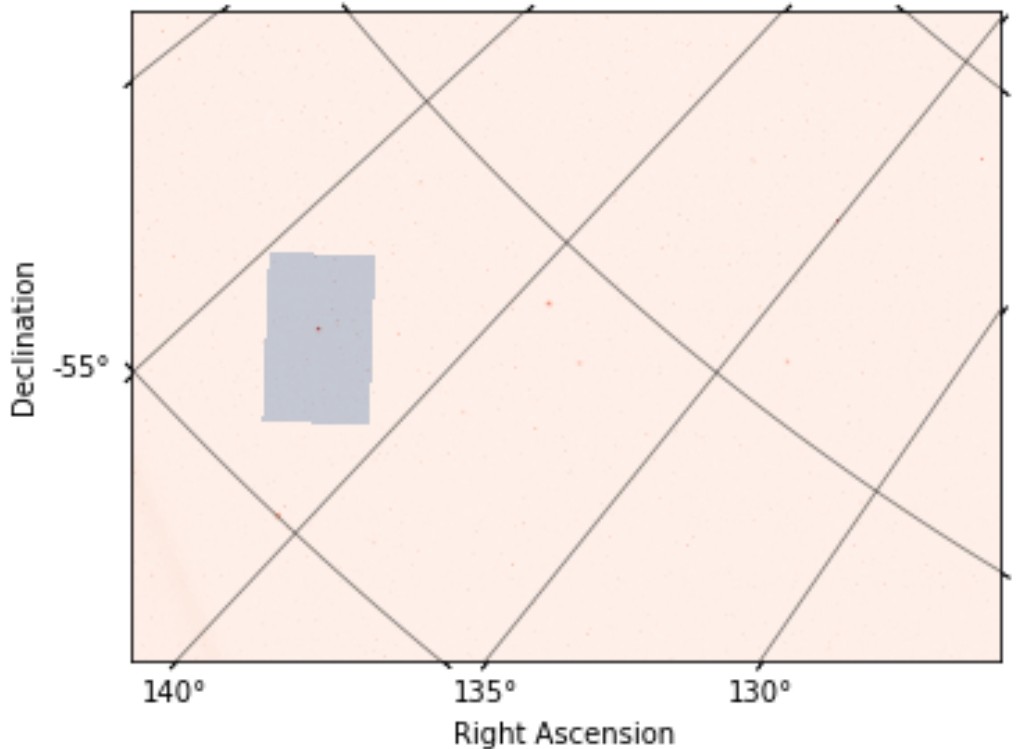

**Figure 3.** GEOStare SV2 image in blue projected to ST view in red

September and October 2020 are referred to as ST data. Observations tasked by Lawrence Livermore National Laboratory with operation by Terran Orbital between November 2021 and April 2022 with a similar nanosatellite and high-resolution telescope are referred to as GEOStare SV2 data (GEOsynchronous Space Telescopes for Actionable Refinement of Ephemeris Space Vehicle 2 – see Simms et al. (2013)). Figure 2 shows an image of the same starfield captured by the telescopes for ST and GEOStare SV2. The smaller GEOStare SV2 field of view is projected onto the star tracker image in Figure 3 to show their 175 respective fields of view (FOV).

ST utilizes a telescope with 14 mm aperture diameter and a 10.7°x 7.8°field of view. The sensor is a MT9M021 CMOS from ON Semiconductor with a 3.75 $\mu$m pixel size for a plate scale of 30.9 arcseconds per pixel operated at a frame rate of 0.5 Hz.

GEOStare SV2 utilizes a telescope with 85 mm aperture diameter and field of view of 2.12°x 1.33°. The sensor is a ASI174MM camera from ZWO with a 5.86 $\mu$m pixel size for a plate scale of 4 arcseconds per pixel operated at a frame 180 rate of 1 Hz.

Figures 2a and 2b show the same starfield captured by ST and GEOStare SV2 with exposure times of 0.284 and 0.5 seconds, respectively. The spatial resolution with GEOStare SV2 is much higher than the ST, as shown by the significantly smaller field of view in the blue image projected to the ST view in red on Figure 3. The telescope utilized by the GEOStare SV2 mission



was designed by Bauman and Pertica (2021); Riot et al. (2017). GEOStare SV2 was designed for space situational awareness
applications while the ST instrument is intended to be a compact, inexpensive, high accuracy star tracker. Neither payload was
designed specifically for stellar occultation, but they were operated to acquire proof-of-concept data to assess our sounding
technique before an instrument designed for stellar occultation is launched.

### 3.1   Star Tracker Observations

Although the ST instrument profiles the atmosphere at a low vertical resolution of 3 - 8 km and captures low resolution starfield
images at 30.7 arcseconds per pixel, the large field of view and long exposure time (0.24 - 0.55 seconds) allows an accurate
estimate of the telescope boresight or WCS, consistent with its intended application as a star tracker. These observations
targeted stars with apparent magnitude 2.35 - 2.54, so each image captured several stars that were unaffected by the atmosphere
or planet in addition to the targeted star. These factors allowed the calculation of an astrometric solution with an uncertainty of
0.3 arcseconds.

We collected data from 217 sessions where each session includes between 15 and 40 images that captured the transit of stars
behind the atmosphere with up to 70 unique stars per session. Not all stars captured are good candidates for stellar occultation
since the star needs to be bright, but not so bright that it saturates the camera. In addition, not all stars are occulted by the
atmosphere during a session. After analyzing the data, we chose the ten brightest occultation events that were not saturated to
analyze.

Figure 4 shows the ST observations and their locations within the Southern subtropics between 26°S and 34°S. Figure 4a and
4b show the full profile and a zoomed in profile of the stratosphere, respectively, with the observed refractive bending angle in
diamond markers from ten sessions. Figure 4b also shows model predictions of the bending angle by applying the Ray tracing
model to the 1976 U.S. Standard Atmosphere (NOAA NASA US Air Force, 1976) in the blue dashed line. The MERRA2
model predicted temperature and density profiles from the Global Modeling and Assimilation Office (GMAO) (2015a, b) for
each occultation session's date and location is used to calculate an estimate of the bending angle, which is shown with the
average profile in the black line and the range among the different times and locations in the grey shaded region. Also, the
mean MERRA2 bending angle at the South Pole and equator averaged over the different times of the occultation sessions are
shown in red dash-dotted and dashed lines, respectively, to illustrate the range of possible stratospheric conditions.

The observed bending angle is calculated from the captured images using the Moffat PSF described in Section 2.2.1 to
determine the centroid locations and the ray perigee is determined with the *direct* method described in Section 2.2.3.

The bending angle observations correlate slightly better with the MERRA2 reanalysis data than the U.S. Standard Atmo-
sphere with correlation coefficients of 0.986 and 0.983, respectively. The parity plot is shown in the Supplemental Information
in Figure B2. The MERRA2 model is much more accurate since it is a global reanalysis model actively developed, so obser-
vations correlating with MERRA2 better than the U.S. Standard Atmosphere provides confidence in these measurements.





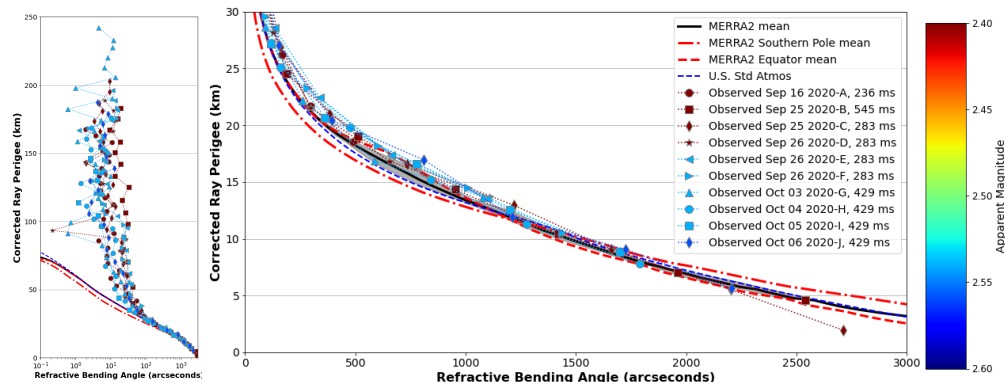

a: Bending angle profiles.

b: Stratospheric bending angle profiles on a linear horizontal scale.

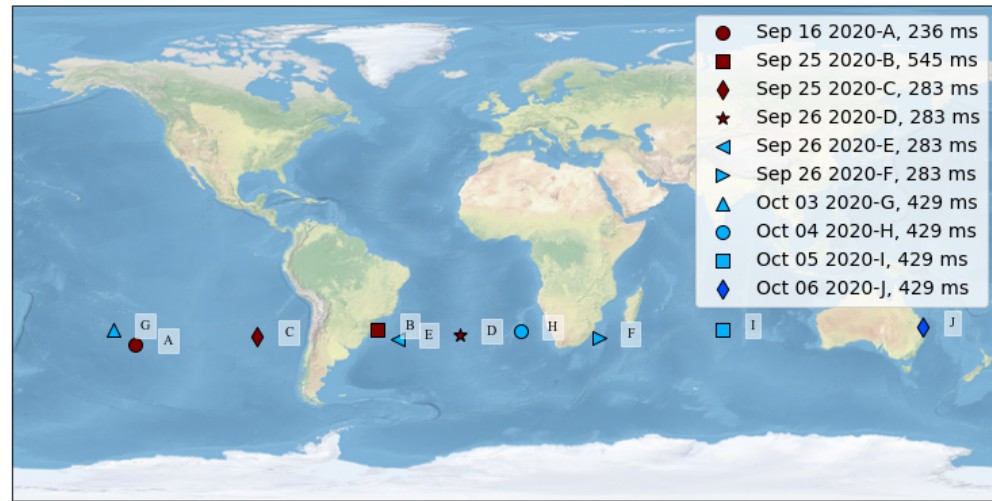

c: Locations of observed profiles.

**Figure 4.** ST observations and their locations in the upper and lower panels, respectively. The top panel shows refractive bending angle profiles observed by ST compared to MERRA2 and U.S. Standard Atmosphere predictions. The gray region is the range of bending angles from MERRA2 across all of our observations. The color of each observation dataset corresponds to the colorbar showing apparent magnitude of the star captured from dimmest (2.6) to brightest (2.4). Figure (a) shows the full atmospheric sounding on a logarithmic scale and (b) shows the profile in the stratosphere on a linear scale. The exposure time of each session's images is listed in the legend entry.



## 3.2 GEOStare SV2 Observations

We scheduled the data collection by pointing at known stars from November 2021 to April 2022 and all of the 48 data sessions captured stars occulted by the atmosphere. We analyze data from 22 of the sessions since they captured a star near the image center that passed across the atmosphere to below 20 km and was not saturated. The images were captured with exposure times between 0.1 ms and 600 ms depending on the target star's apparent magnitude.

The small field of view and short exposure times do not capture enough objects in the starfield to accurately determine an astrometric solution for every image. Since we need to account for the WCS of each image, we utilized data on the payload attitude and location from the onboard star trackers to calculate the image boresight and rotation. This attitude-derived WCS provides an estimate of the motion of the telescope throughout a session with an uncertainty of 12 arcseconds.

Figure 5 shows the location and observed refractive bending angle for the 22 sessions, which profiled a wide range of locations across the globe. Figure 5a and 5b shows the full profile and a zoomed in stratospheric profile of the GEOStare SV2 observations, respectively. The average location of the ray perigee on the Earth surface is shown in Figure 5c as circles labeled A - V with the same color as the observations shown in the top panels.

The bending angle profile is calculated with star centroids fit to a 2D Gaussian PSF and the ray perigee calculated with the *rotated* method described in Sections 2.2.1 and 2.2.3, respectively. Point sources captured by the optics in GEOStare SV2 match a diffraction-limited spot better than the ST optics, so a Gaussian function is a better match for the centroid shape. On the other hand, the uncertainty in image WCS makes calculating the direct ray perigee difficult, so the rotated calculation is used with a reference coordinate taken from the catalog position of the target star.

The data captured here samples the profile more frequently than ST, but the instrument error appears significantly higher since the observed bending angle deviates from the MERRA2 and U.S. Standard Atmosphere predictions up to altitudes of 30 km. All sessions are shown in Figure 5b with dotted lines. We highlight four specific sessions on Feb 19, 20, 21, and 28, differentiating the latter three with different line styles. During the Feb 19, 20, and 21 sessions shown in the blue dotted, solid, and dashed lines, respectively, the telescope orientation significantly rotated. The rotation is captured with the payload attitude data, but the frequency of the attitude control system data does not necessarily align with the sampling rate or timing of the captured images and there is significant uncertainty associated with the attitude-derived WCS. The large camera rotation likely caused the significant deviation of the observed bending angle profile from the models. The Feb 28 session shown in a solid line successfully observed the atmospheric bending angle at low as 3 km. This session targeted the bright star with apparent magnitude 0.98, Aldebaran, when there were presumably no clouds or objects blocking the star wavefront.

## 3.3 Retrieved Temperature

As shown in Figure 4a, data collected above 40 km is at too low of a signal-to-noise ratio (SNR) to gather any atmospheric information. Since the retrieval at an altitude $z$ integrates all observations above $z$, we cut-off the observed bending angle at $z_{max}$ to avoid corrupting the retrieval with too much instrument noise. Figures 6a, 6b, and 6c show the final retrieved



temperature profile from ST with $z_{max}$ of 30 km, 40 km, and 60 km, respectively. Similarly, Figures 6d, 6e, and 6f show the final retrieved temperature profile from GEOStare SV2 with $z_{max}$ of 20 km, 30 km, and 50 km, respectively.

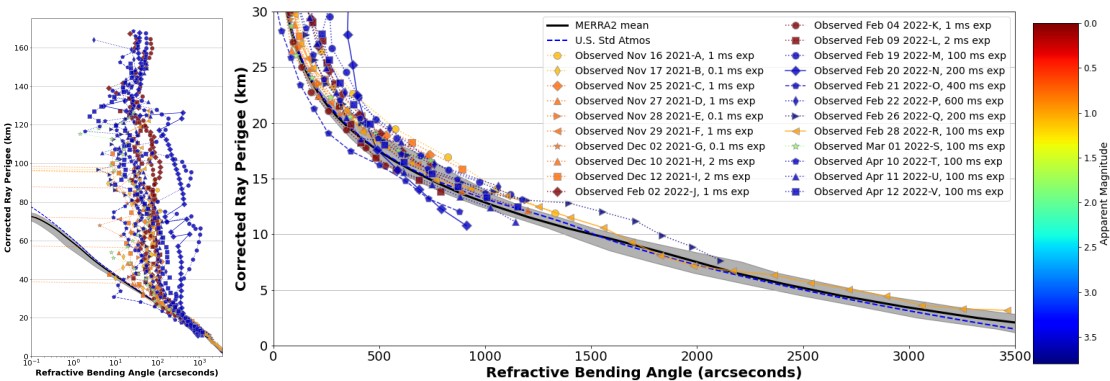

a: Bending angle profiles.

b: Stratospheric bending angle profiles on a linear horizontal scale.

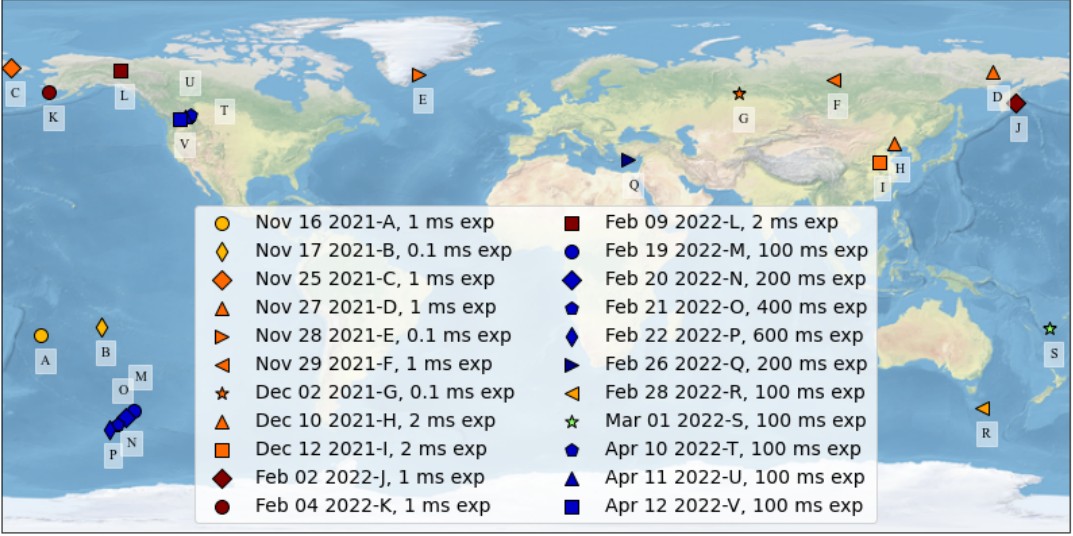

c: Locations of observed profiles.

**Figure 5.** GEOStare SV2 observations and their locations in the upper and lower panels, respectively. See Figure 4 for details. Here the stars captured range from dim to bright with apparent magnitudes 3.6 to -0.04.





The top and bottom middle figures in Figure 6 show the best $z_{max}$ for each instrument: 40 km and 30 km for ST and
GEOStare SV2, respectively. The left and right figures show cases with too much and too little data removed, respectively.

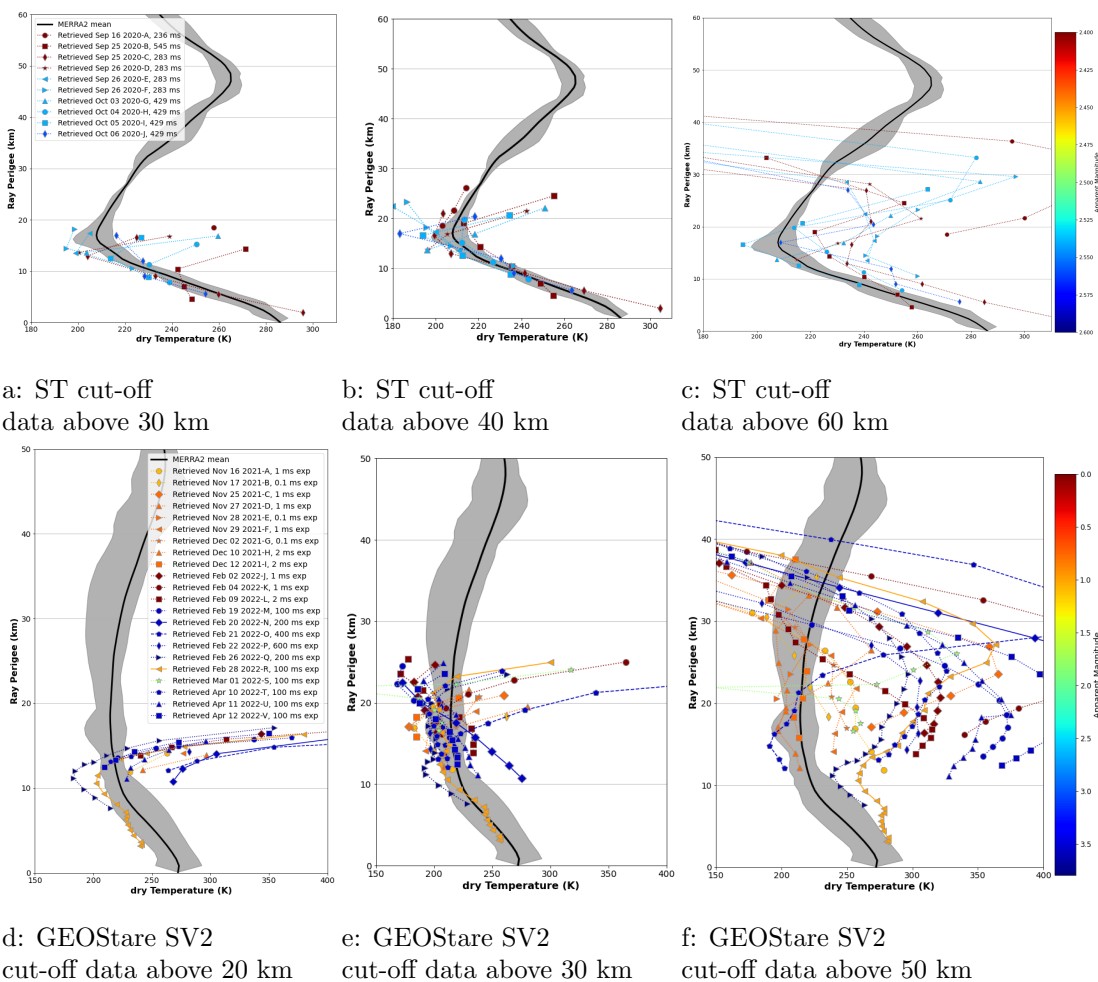

a: ST cut-off
data above 30 km

b: ST cut-off
data above 40 km

c: ST cut-off
data above 60 km

d: GEOStare SV2
cut-off data above 20 km

e: GEOStare SV2
cut-off data above 30 km

f: GEOStare SV2
cut-off data above 50 km

**Figure 6.** Retrieved temperature profiles from ST and GEOStare SV2 observations. The observation marker shape and color corresponds to
Figures 4 and 5 according to the captured star's apparent magnitude shown in the colorbars on the right for each measurement dataset. The
mean temperature profile predicted by MERRA2 is shown in the black line with the range across the varying measurement locations and
dates shown in the gray shaded area.



A low cut-off altitude results in too few observations for an accurate retrieval, while a high $z_{max}$ incorporates too much instrument error in the retrieval. However, the retrieved temperature profile is accurate at altitudes below 20 km for both of these instruments with the exception of a few datasets (Feb 20 & 21) that were collected with a rotating field of view.

## 4 Sounding Technique Analysis

To assess the proposed atmospheric sounding technique, we utilize the refraction forward model described in Section 2.1 based on the Naval Research Laboratory empirical atmospheric model NRLMSISE0 (Emmert et al., 2021) over the Pacific Ocean. The modeled refractive bending angle is sampled at 500 meter vertical resolution and noise is added to simulate satellite measurements. Then, the retrieval technique described in Section 2.3 is applied to the simulated noisy measurements and compared with the NRLMSISE0 temperature profile to analyze how well the sounding technique can estimate the stratospheric

temperature profile.

### 4.1 Measurement Error

Measurement error in the bending angle is due to uncertainty in the centroid position since it is calculated from the distance between the observed star centroid and its reference centroid position. The centroid position uncertainty $\sigma_x$ in pixels is characterized with uncertainty due to background signal $\sigma_{bkd}$, photons captured from the source signal $\sigma_{signal}$, atmospheric

turbulence $\sigma_{turb}$, and boresight pointing $\sigma_{BS}$ (Thomas et al., 2006).

$$\sigma_x = \sqrt{\sigma_{bkd}^2 + \sigma_{signal}^2 + \sigma_{turb}^2 + \sigma_{BS}^2} \tag{11}$$

The centroid error due to background signal depends on the spot size described by its full width half-maximum ($FWHM$), the spot strength based on the number of photons captured $N_{ph}$, and the background strength in terms of the number of photons across the spot $N_{bkd}$. Any centroid blurring due to jitter of the spacecraft is captured by the estimated spot size $FWHM$. The

270 centroid signal $N_{ph}$ is based on the star apparent magnitude, the exposure time, the efficiency and transmission of the optical system, and transmission loss due to Rayleigh scattering. The background signal of the sky and airglow is estimated from the measured intensity in ADU/(s Pixel), the camera inverse gain in electrons/ADU, the exposure time, and the spot size.

The centroid uncertainty $\sigma_{signal}$ increases as the source signal and $N_{samp}$ decreases. $N_{samp}$ represents the number of pixels sampled by a diffraction-limited spot based on the plate scale $p$, light wavelength $\lambda$, and the aperture diameter $D_{aper}$.

Turbulence impacts the centroid by blurring the spot during the camera integration timeframe if the turbulence is strong enough. The turbulence strength is characterized by the Fried parameter $r_0$ cm (Fried, 1965), which is defined as the required aperture diameter to resolve atmospheric turbulence. Perlot (2009) describes the 5/7 Hufnagel-Valley model ($HV_{5/7}$) we use for a typical atmospheric turbulence vertical profile. Thomas et al. (2006) describes the effect of turbulence based on an empirical parameter $K$, the window size, Fried parameter, and aperture diameter.

Uncertainty in the boresight pointing directly propagates to the centroid uncertainty with $\sigma_{BS}$.



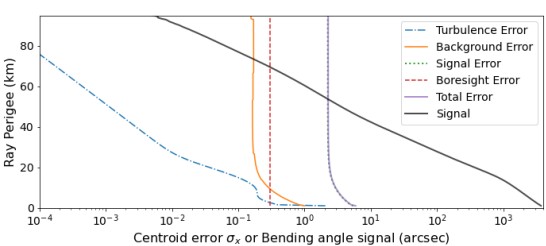
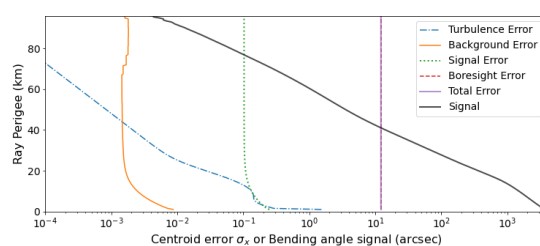

a: ST bending angle error profile        b: GEOStare SV2 bending angle error profile

**Figure 7.** Bending angle error analysis for each satellite. The thick, solid black line shows the expected bending angle. The blue dash-dotted, orange solid, green dotted, and red dashed lines show the uncertainty in centroid position due to atmospheric turbulence, background signal, photon signal and centroiding, and satellite pointing, respectively. The total uncertainty in the centroid position is shown in the purple solid line.

Based on the details of each satellite and telescope configuration, we estimate the measurement error vertical profile. The equations and parameters used for each instrument are described in the Appendix and in Section 3 assuming each set-up utilizes one of the dimmer stars captured during the actual data collections corresponding to Markeb and Algedi for ST and GEOStare SV2, respectively. This corresponds to a 2.5 and 3.57 apparent magnitude star, respectively, and integration time
of 430 and 100 ms, respectively. Applying the details of each instrument to the centroid uncertainty, the bending angle error and its components are shown in Figure 7 for ST and GEOStare SV2 along with the expected bending angle, or measurement signal, for reference.

The signal and background errors depend on $N_{ph}$, the star brightness, and as light passes deeper into the atmosphere there is transmission loss due to Rayleigh scattering that dims the observed star. Therefore, these sources of uncertainty grow towards
the Earth surface in the lower stratosphere and troposphere. The background signal also slightly increases near the airglow at 80-100 km and below as the sky brightness increases. GEOStare SV2 has a much lower background signal due to its design to keep stray light from the telescope. This analysis shows that the star brightness and uncertainty in the boresight are the most significant sources of uncertainty in ST and GEOStare SV2 data, respectively.

## 4.2 Retrieval Analysis

To evaluate our retrieval technique, we simulate noisy satellite measurements by corrupting the modeled bending angle sampled at 0.5 km resolution with white noise scaled by the centroid uncertainty profile. Then, the retrieval method described in Section





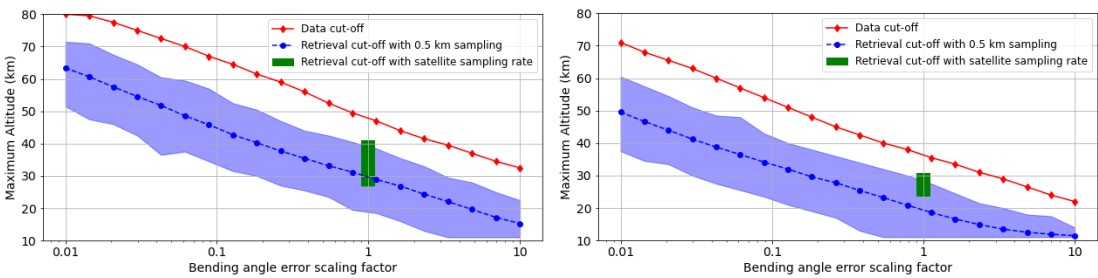

a: ST Retrieval Sensitvitity      b: GEOStare SV2 Retrieval Sensitvitity

**Figure 8.** Maximum altitude of retrieved atmospheric sounding profile analysis for each satellite measurement. The red diamonds show the data cut-off due to a degraded SNR relative to the amount of added bending angle noise. The maximum altitude according to a threshold of 2% error in retrieved temperature is shown with the mean and range of the ensemble in blue circles and blue shaded area, respectively. The maximum altitude of 10% error in retrieved temperature utilizing the observed vertical resolution is shown in the green shaded range for the lowest to highest vertical resolution of each instrument.

2.3 is applied to the section of simulated bending angle measurements with at least a Signal to Noise Ratio (SNR) of two. The altitude corresponding to a SNR of two is the data cut-off altitude. However, there is still error in the retrieved temperature compared to the true temperature profile near the upper and lower sections of the profile, even with a SNR above two. Therefore, we assess the retrieval cut-off altitude by determining the level (above 10 km) below which the percent error is below 2%, so the retrieved temperature error is below a 2% threshold between the retrieval cut-off and 10 km. 2% is the threshold for being precise enough to detect atmospheric gravity waves.

The above retrieval analysis is performed for the instrument nominal case and several other cases with the bending angle error scaled by a factor from 0.01 to 10. For each bending angle error scaling factor, we simulated 1000 realizations of the noisy measurements calculating the cut-off altitudes for each scenario, and the average result is shown here. The data cut-off $z_{max}$ and mean retrieval cut-off is shown in Figure 8 with solid red diamonds and dashed blue circles, respectively. Additionally, the range in the retrieval cut-off among the ensemble of 1000 scenarios is shown in the blue shaded region. The result for ST data is shown in Figure 8a with higher cut-off altitudes than GEOStare SV2 data shown in Figure 8b due to its lower nominal bending angle error.

This analysis shows theoretical performance of each instrument across limiting cases (1) if aspects of the measurements are improved resulting in a factor of 100 lower bending angle error to (2) if the measurement is degraded so the error is enhanced by a factor of 10. The worst-case scenario simulated with GEOStare SV2 set-up and the bending angle error degraded by





a factor of 10 results in a noise floor of 121 arcseconds. In this case, 55% of the noisy measurement realizations provide retrieved temperature within the error threshold of 2% and the remainder only retrieve up to 11.5 km on average. In contrast, with a scaling factor of 0.03 on the ST instrument, the mean bending angle error is similar to the current background error approximately 0.07 arcseconds. This scenario results in retrieved temperature from 10 km up to 42 – 64 km with an average maximum altitude of 55 km.

This analysis does not reproduce retrievals from the satellite measurements since the atmosphere was sampled at a frequency of 0.5 km and the real observations were sampled with spacing between 1.5 and 8 km. The lower sampling rate results in poor quality retrievals with the Abel inversion technique, so to assess the real observation system we use a threshold of 10% temperature error instead of 2%. The data cut-off for the lower sampling rate is approximately the same as the higher sampling rate shown in the red diamond at a scaling factor of one in Figure 8. The range in retrieval cut-off for a sampling rate between 3 - 8 km and 1.5 - 6.5 km for ST and GEOStare SV2, respectively, are shown in green ranges on Figure 8.

## 5  Future Observations

A future instrument, Stellar Occultation Hypertemporal Imaging Payload (SOHIP), is planned to launch in January 2023 joining the suite of instruments on the International Space Station. SOHIP can be leveraged to achieve a higher SNR since it includes the high resolution camera of GEOStare SV2 with an even higher frame rate along with a second telescope somewhat similar in size to a Terran Orbital star tracker. Operating two telescopes in tandem will mitigate the tradeoff discussed above, between $\sigma_{BS}$ and $\sigma_{signal}$, when choosing an integration time.

The star tracker camera onboard SOHIP can capture a large field of view with a long integration time to reduce the bore-sight error from $\sigma_{BS}$=12 arcseconds observed with GEOStare SV2 to $\sigma_{BS}$=0.3 arcseconds when determining an astrometric solution. The high resolution, "science" telescope onboard SOHIP is the same design and specifications as the GEOStare SV2 telescope, so $\sigma_{bkd}$ is below 0.01 arcseconds. The goal of SOHIP is to observe a high vertical resolution profile with the stellar occultation methodology, so the science telescope will operate with a high frame rate near 1 kHz and thus a short integration time. This limits the targets to only the brightest stars, and the $\sigma_{signal}$ is 0.2 arcseconds similar to GEOStare SV2 due to the short integration.

This overall design results in a more accurate bending angle measurement with $\sigma_x$ of 0.39 arcseconds. This noise floor corresponds to a maximum retrieval altitude between 30 and 50 km based on a 2% error threshold in temperature (41 km maximum altitude on average). Additionally, the accuracy and precision of retrieved temperature at 25 km is predicted to be 0.5 and 0.7 K, respectively (not shown).

## 6  Conclusions

The technique and observations presented here serve as proof-of-concept for utilizing nanosatellites for stellar occultation measurements for atmospheric sounding. The technique presented here captures an optical wavefront as the starlight passes through





the atmosphere, adjusting the apparent centroid position based on the atmospheric refractivity. We observe a vertical profile of
the atmospheric refractive bending angle, which allows us to retrieve atmospheric refractivity, density, and temperature profiles.

We present data gathered from two types of on orbit sensors, a dedicated imaging telescope on the GEOStare SV2 mission, and data collected from ST sensors. Both instruments result in significant uncertainty so that the highest altitude of observed bending angle is 30 km. ST observations are limited by the relatively low-resolution sensor, but the long exposure and large field of view enabled an accurate estimate of the telescope pointing coordinates and orientation. On the other hand, GEOStare
SV2 is limited by the error in attitude-derived telescope pointing, not due to satellite stability but the mode of data collection with short integration times even though the instrument has a high resolution camera and sampling rate.

Simulated measurements based on the known source of uncertainty contributing to the bending angle agree with the observed signal-to-noise ratio (SNR). We find that the profile can be extended from an upper limit of 20 km if the overall error is decreased by a factor of 100 for ST or GEOStare SV2 up to 60 km or 50 km, respectively, assuming 2% error in retrieved
temperature. An upcoming 2023 SOHIP mission is expected to retrieve temperature up to 41 km on average and a precision of 0.7 K at 25 km with a high vertical resolution near 3 m. This is due to utilizing both a star tracker and GEOStare SV2 sensor to detect atmospheric gravity waves.

This technique is a promising, inexpensive method to observe the stratospheric temperature profile and potentially observe small-scale perturbations in the atmospheric field. The small optical wavelengths captured with stellar occultation allow the
observation of small-scale upper atmosphere phenomena like atmospheric gravity waves and turbulence. This method can lead to a remote observation method able to probe sections of the atmosphere where turbulence currently cannot be detected or measured, which is relevant for future supersonic airliners. A future instrument, SOHIP, aims to achieve this by utilizing two cameras capturing a bright star with both a long and short exposure time providing high SNR bending angle observations.

## Appendix A: Instrument Error

Definitions and physical constants used:

- $t_{exp}$ exposure time (s)

- $m$ apparent magnitude of star

- $I_{bkd}(z)$ intensity of background signal in night sky at ray perigee $z$ (ADU/pixel-s)

- $\mathcal{G}$ inverse gain of detector (e$^-$/ADU)

- $FWHM \cong 2$ pixels - full width half max of source

- $N_{pix} \cong FWHM^2$ number of pixels subtended by star

- $N_{samp}$ half-width of diffraction-limited spot (pixels)

- $r_n$ electrons of read noise ; $N_r^2$ read out noise variance





- $p$ plate scale in pixel per radian

- $\sigma_x$ uncertainty in source position in arcseconds

- $A_{aper}$ Aperture area from diameter $D_{aper} = 8.5$ cm

- $Q_e$ & $f_{opt}$ Quantum efficiency and transmission of optical system (0.7 & 0.9, resp)

- $BW$ bandwidth (3000 Å)

- $F_0$ Vega photon flux in R-band (600 $\mathrm{photon\,cm^{-2}s^{-1}}^{-1}$)

- $f_{scat}(z)$ fraction of signal transmitted (due to Rayleigh scattering at 400 nm wavelength) at ray perigee $z$

- $\lambda$ wavelength (0.7 $\mu$m)

- $K$ coefficient for turbulence effect on centroid (0.5 pixel$^2$)

- $W_{pix} := FWHM$ window length (2 pixels)

- $r_0$ Fried parameter

- $\sigma_{BS}$ uncertainty in boresight pointing (pixels)

- $\delta$ FWHM of Gaussian spot (pixels)

The centroid uncertainty $\sigma_x$ is determined with the following equations.

$$\delta = FWHM/\sqrt{2} \tag{A1}$$

$$N_{ph} = F_0 BW A_{aper} 2.512^{-m} t_{exp} Q_e f_{opt} f_{scat}(z) \tag{A2}$$

$$N_{samp} = \frac{p\lambda}{D_{aper}} \tag{A3}$$

$$N_r = \sqrt{N_{pix} r_n^2} \tag{A4}$$

$$N_{bkd} = \sqrt{\mathcal{G} I_{bkd}(z) N_{pix} t_{exp}} \tag{A5}$$

$$r_0(z) = \left[ 0.423(2\pi/\lambda)^2 \int\limits_{z}^{100} \sec\xi(h)\mathrm{d}h C_N^2(h) \right]^{-3/5} \tag{A6}$$

$$\sigma_x = p^{-1} \sqrt{\frac{4\delta^2 N_{bkd}^2}{N_{ph}^2} + \frac{\pi^2}{2ln(2)N_{ph}} \frac{FWHM^2}{N_{samp}^2} + K\left(W_{pix}\frac{D_{aper}}{p\lambda}\right)^{-2}\left(\frac{D_{aper}}{r_0(z)}\right)^{5/3} + \sigma_{BS}^2} \tag{A7}$$



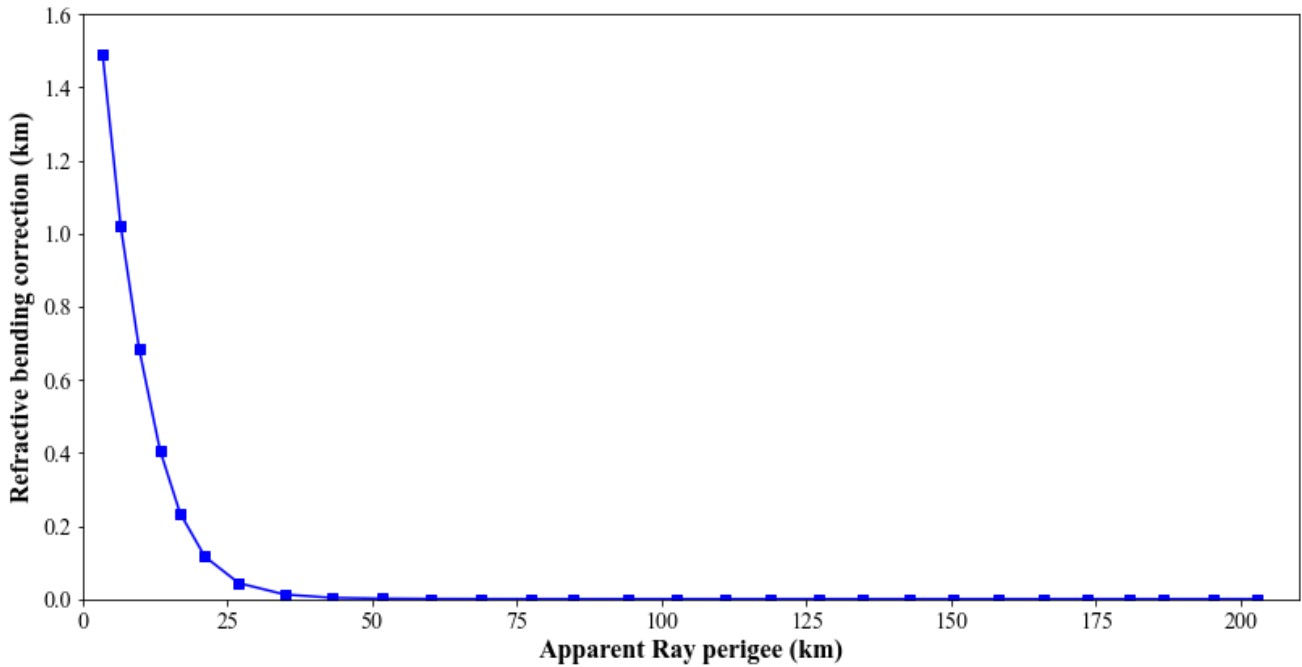

**Figure B1.** Correction to ray perigee to account for refractive bending with ray tracing algorithm.

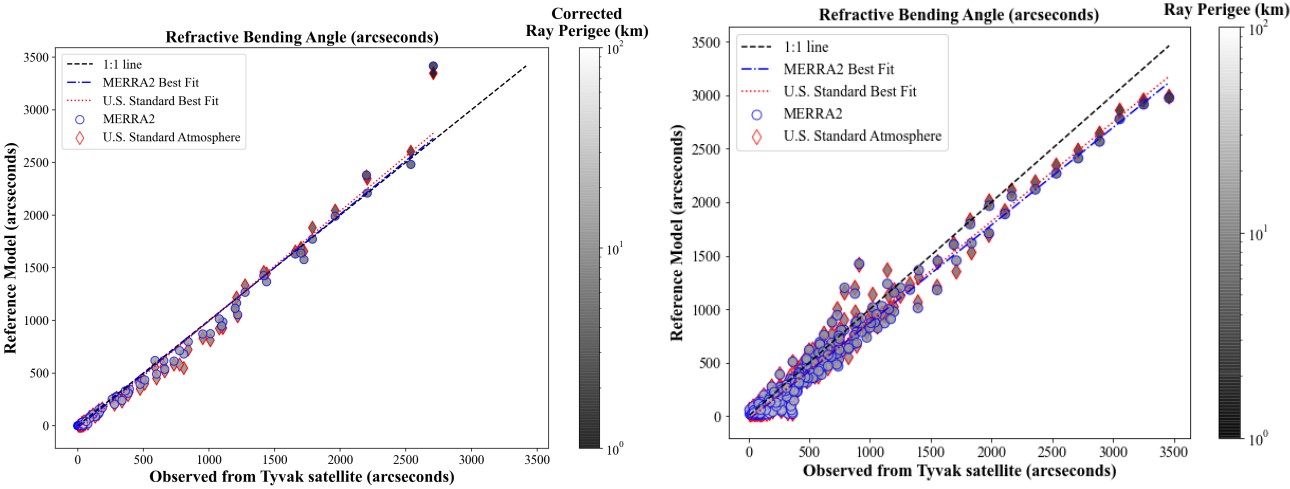

**Figure B2.** Parity plot of the U.S. Standard and MERRA2 reference model predictions against the observed bending angle from star tracker and GEOStare SV2 on the left and right plots, respectively. The ray perigee of each point is shown in the grayscale colorbar.


## Appendix B: Supplemental Figures

The observed bending angle by star tracker matches the MERRA2 model better than the U.S. Standard Atmosphere, as shown in Figure B2 with a parity plot between the observations and the two reference models. The blue circles represent the comparison of the data with MERRA2 while the red diamonds show the comparison with U.S. Standard Atmosphere resulting in correlation coefficients of 0.986 and 0.983, respectively. The MERRA2 model is much more accurate since it is a global reanalysis model actively developed, so observations correlating with MERRA2 better than the U.S. Standard provides confidence in these measurements.

*Data availability.* Data can be made available upon request.

*Author contributions.* Conceptualization by M.H., P.CS., D.M., B.B., L.S.; data collection by C.S., D.H., A.P., W.D.; analysis by D.M., A.P., W.D., M.H., P.CS.; manuscript prepared by D.M. with direct contributions from P.CS., C.S., W.D., B.B., L.S., M.H., A.P..

*Competing interests.* The authors declare the following financial interests/personal relationships which may be considered as potential competing interests:

Brian Bauman has patents #10935780 and #9720223 licensed to Lawrence Livermore National Laboratory.

*Acknowledgements.* This work was performed under the auspices of the U.S. Department of Energy by Lawrence Livermore National Laboratory under Contract DE-AC52-07NA27344 with document number LLNL-JRNL-840198. This work was funded by Laboratory Directed Research and Development Exploratory Research project "Observing Atmospheric Gravity Waves from Space" with tracking code 20-ERD-007. GEOstare SV2 is operated by Terran Orbital and LLNL under a Collaborative R&D Agreement (CRADA #TC02342). The AGI Product Support team provided invaluable help and guidance in using the STK software for our calculations.





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
