# Peer review of "Stratospheric Temperature Measurements from NanoSatellite Stellar Occultation Observations of Refractive Bending"

_Atmospheric Measurement Techniques, 2022_

## Author Response (AR1)

**1. Reply to RC1**

Thank you for the positive feedback and comments that improved this paper. Please see our response to each comment below each point in **bold underlined text.**

1) Introduction: Please quote Sofieva et al. (2019) who recover temperature profiles from bichromatic stellar scintillations using GOMOS observations.

- **Citation to Sofieva et al. (2019) is now included on line 37.**

2) Line 92: the highest pressure level predicted by MERRA2 is 0.01 hPa, which corresponds to about 80 km, not 86 km.

- **This is corrected on line 93.**

3) Paragraph 2.2.1: The PSF of the centroid of the star is descrobed by a Moffat function. How do you determine the parameters included in this distribution (width and negative exponent) and what are their values?

- **The values varied for each centroid, so we added the range in fitted width and negative exponent parameters to line 116. These values are determined from a least-square fit, which is now described in more detail on line 107.**

4) Line 154: n cannot be neglected in the calculation of the impact parameter p near the ground. For n=1.003, the error on p will be 1.9 km, which is not negligible.

- **This is a good point for any measurement at ray perigee of 1 km, which is much lower than the measurements presented in this work. However, the error at 10 km is below 10% (only 0.61 km). A more detailed treatment would include this effect on the impact parameter within the retrieval, but we will neglect n in this work since most of the data is above 10 km. Since we do present limited retrieval results at altitudes below 10 km, we now include a caveat highlighting this on line 259. Additionally, we justify the simplification based on the refractive index at 10 km instead of 1 km on line 157.**

5) Lines 287-288: The brightness of the star is also attenuated due to refractive dilution due to the decrease in refractive index with altitude as explained in Sofieva et al. 2007.

- **This is a good suggestion and is now included in the uncertainty calculations. The attenuation fraction (q) is applied to the star photon signal (Nph) in Eq A2, and this is described in the text on line 279. The results are updated including the effect of refractive dilution, but they did not significantly change since the signal to noise ratio is**

greater than 10 at low altitudes where the dilution becomes important. Thus, we did not need to update any of our conclusions or discussion.

**2. Reply to RC2**

Thank you for the positive feedback and comments that improved this paper. Please see our response to each comment below each point **in bold underlined text**.

Title: The term "Observations of Stellar Occultation Bending" is very uncommon. I suggest "Stellar Occultation Measurements of Refractive Bending" or similar.

- **We agree this title is clearer, so the updated title is: Stratospheric Temperature Measurements from NanoSatellite Stellar Occultation Observations of Refractive Bending**

Abstract: Line 2: What do you mean by "unlike other measurement techniques like radiosondes, aircraft, and radio occultation." These techniques can provide fine vertical profiles of atmospheric temperature too. Please clarify.

- **This statement aimed to highlight the advantages of this measurement technique are its ability to measure fine-scale phenomena (unlike radio occultation) and its potential to observe altitudes above 30 km (unlike radiosondes and aircraft). This statement is now clarified in the abstract stating:**
- **"Stellar occultation observations from space can probe the stratosphere and mesosphere at a fine vertical scale around the globe. Unlike other measurement techniques like radiosondes and aircraft, stellar occultation has the potential to observe the atmosphere above 30 km and unlike radio occultation, stellar occultation probes fine-scale phenomena with potential to observe atmospheric turbulence."**

Figure 2: Based on Declination and RA it seems that there is no overlapping area in both images. Adding a label for the brightest stars in both images would help to identify common points. In addition, Figure 2b requires a proper y-labeling, from -58° alone the scale of the image cannot be identified. Further suggestion: The external boarder of Figure 2b could be added to Figure 2a so that the overlapping area becomes clear.

- **We found a better way to visualize the starfield images on the skyfield world coordinates with the additional axis labels and rotated the Figure 2b image so it is on the same orientation coordinates as Figure 2a. The border of Fig 2b image is drawn in red on Fig 2a to further guide the reader.**

Figure 3: With the modifications suggested for Figure 2, Figure 3 is not necessary anymore.

- **Figure 3 has been removed.**

Line 189: How did you come up with a 3 – 8 km vertical resolution for the ST instrument?

- **This number came from the range in change in altitude between sequential frames after processing the data to determine the ray perigee of each frame. The text has been updated to clarify this point:**
- **Line 192: "Although the ST instrument profiles the atmosphere at a low frequency of 0.5 Hz and captures low resolution..."**
- **Line 202: "We found the vertical resolution ranges from 3 km to 8 km among these ten occultation events."**

Line 200: Is there a special reason why ST observations were selected within the latitude range 26-34°S only?

- **The latitude of observations was not considered in the data collection strategy, but it likely ended up in a small range of latitudes due to a combination of many factors. Some causes of the small latitude ranges include: the camera operation limitation (stars brighter than 2.4 apparent magnitude are saturated), our filtering of the dataset for the brightest stars, the satellite orbit characteristics, and the viewing direction of the star tracker camera. Combination of the camera operation limitation and data filtering resulted in observing the same star occulted in several instances. Targeting the same star with a limited range of motion in the camera field of view, it is reasonable that the observed latitude range is small. We clarified this with:**
- **Line 204: "The observations were not aimed at a particular geographic location, so the combination of the camera operation limiting the brightness of stars, filtering data for the brightest stars, and orientation of the star tracker camera led to unintentionally targeting the Southern subtropics."**

Figure 4a,b, 5a,b and Figure 6: Labels and legend are difficult to read. Increase font size.

- **The font sizes are now larger. We additionally adjusted the color scales to be colorblind-friendly.**

Line 343: The first two sentences of the Conclusions start with "The technique … presented here .." Please rephrase.

- **The second sentence is rephrased to start with "Stellar occultation captures ..."**

Not sure if a vertical resolution of 3 m is possible given the expected diffraction-limit?

- **We have clarified that the 3 m quantity is actually the potential vertical sampling, which hopefully answers this question. We reworded the "high vertical resolution near 3 m" to say "high vertical sampling near 3 m" on line 366. Additional explanation is added to the section on Future Observations on line 345.**